# Feasibility of fresh frozen human cadavers as a research and training model for endovascular image guided interventions

**Marloes M. Jansen** [1]*, **Constantijn E. V. B. Hazenberg**[1], **Quirina M. B. de Ruiter**[1], **Robbert W. van Hamersvelt**[2], **Ronald L. A. W. Bleys**[3], **Joost A. van Herwaarden**[1]

**1** Department of Surgery, University Medical Center Utrecht, Utrecht, The Netherlands, **2** Department of Radiology, University Medical Center Utrecht, Utrecht, The Netherlands, **3** Department of Anatomy, University Medical Center Utrecht, Utrecht, The Netherlands

* m.m.jansen-38@umcutrecht.nl

## Abstract

### Objective

To describe the feasibility of a fresh frozen human cadaver model for research and training of endovascular image guided procedures in the aorta and lower extremity.

### Methods

The cadaver model was constructed in fresh frozen human cadaver torsos and lower extremities. Endovascular access was acquired by inserting a sheath in the femoral artery. The arterial segment of the specimen was restricted by ligation of collateral arteries and, in the torsos, clamping of the contralateral femoral artery and balloon occlusion of the supra-truncal aorta. Tap water was administered through the sheath to create sufficient intraluminal pressure to manipulate devices and acquire digital subtraction angiography (DSA). Endovascular cannulation tasks of the visceral arteries (torso) or the peripheral arteries (lower extremities) were performed to assess the vascular patency of the model. Feasibility of this model is based on our institute's experiences throughout the use of six fresh frozen human cadaver torsos and 22 lower extremities.

### Results

Endovascular simulation in the aortic and peripheral vasculature was achieved using this human cadaver model. Acquisition of DSA images was feasible in both the torsos and the lower extremities. Approximately 84 of the 90 target vessels (93.3%) were patent, the remaining six vessels showed signs of calcified steno-occlusive disease.

### Conclusions

Fresh frozen human cadavers provide a feasible simulation model for aortic and peripheral endovascular interventions, and can potentially reduce the need for animal experimentation. This model is suitable for the evaluation of new endovascular devices and techniques or to master endovascular skills.

**Data Availability Statement:** All data is held in a public repository. All files are available from the Mendeley Data and can be found as: Jansen,

Marloes (2020), "Fresh frozen human cadavers for endovascular simulation | Experimental data", Mendeley Data, V1, doi: 10.17632/jnj76frh5y.1.

**Funding:** The author(s) received no specific funding for this work.

**Competing interests:** The authors have declared that no competing interests exist.

# Introduction

Rapid succession of catheter-based innovations in the last decade, has led to more sophisticated endovascular procedures and increased patient eligibility for endovascular treatment [1–4]. Preclinical models are essential to evaluate the feasibility, safety, and efficacy of any novel endovascular device or technique, and to obtain regulatory approval for their clinical application [5, 6]. Preclinical models are also used in endovascular training and device instructions to stimulate safe and efficient implementation of novelties in clinical practice.

Various types of preclinical models are available for endovascular research and training purposes, each with its own advantages and disadvantages. Bench models, such as silicone aorta phantoms, provide a highly controlled environment but lack the complexity and tissue properties of clinical practice. Animal models, such as the well-established healthy swine model, allow in-vivo endovascular simulation, but the healthy, straight arteries of the swine differ considerably from the tortuous, calcified vessels of a typical vascular patient. Besides, the vasculature of the swine is narrower than that of humans, which can preclude the use of larger sheaths and devices. These anatomical and pathophysiological distinctions can compromise the translation from bench-to-bedside [7–10]. Moreover, there are strong ethical grounds to look for alternatives for the use of living animals. Human cadaver reperfusion models have been reported as a suitable, although less familiar, alternative to animal models for endovascular device evaluation and training [11–14]. However, reperfusion of cadaveric tissue is a delicate and complex process, which may restrict widespread adoption of human cadaver models.

In contrast to these human cadaver reperfusion models, our institution has been using human cadaver models without arterial reperfusion, to our satisfaction. In this study, we present our human cadaver model, and aim to raise general awareness of the existence and feasibility of human cadaver models for endovascular simulation in the abdominal aorta and peripheral arteries.

# Materials and methods

## Human cadavers

Body donations were acquired directly by our institute's anatomy department. Body donations were regulated according to the national Dutch law on the disposal of the dead and the European legislations and ethical framework for body donation [15]. Written informed consent was obtained from the donors during life to use their cadavers for educational and research purposes. No additional approval of an ethics board was required for this study. Age, gender and serology report of the donors were provided. Other demographics and medical details were sealed to assure the anonymity of the donors, as according to our institution's policy.

The available data of 22 lower extremities and 6 fresh frozen human cadaveric torsos were used for the assessment of this cadaver model. Donors were predominantly male (75%) with a mean age of 74.4 ± 8.8.

## Cadaveric specimen

The cadavers were dissected and frozen within 48 hours of post-mortem delay, at a temperature of -20˚C. No additive preservation chemicals were used. The cadaveric torsos comprised the thorax, abdomen, pelvis and proximal part of the lower extremities until mid-femoral level. Two oblique dissection planes from axilla to sternum exposed the branches of the aortic arch, trachea, esophagus and cervical spine. Two axial dissection planes at mid-femoral level exposed the left and right femoral artery. The lower extremities comprised the foot, lower leg

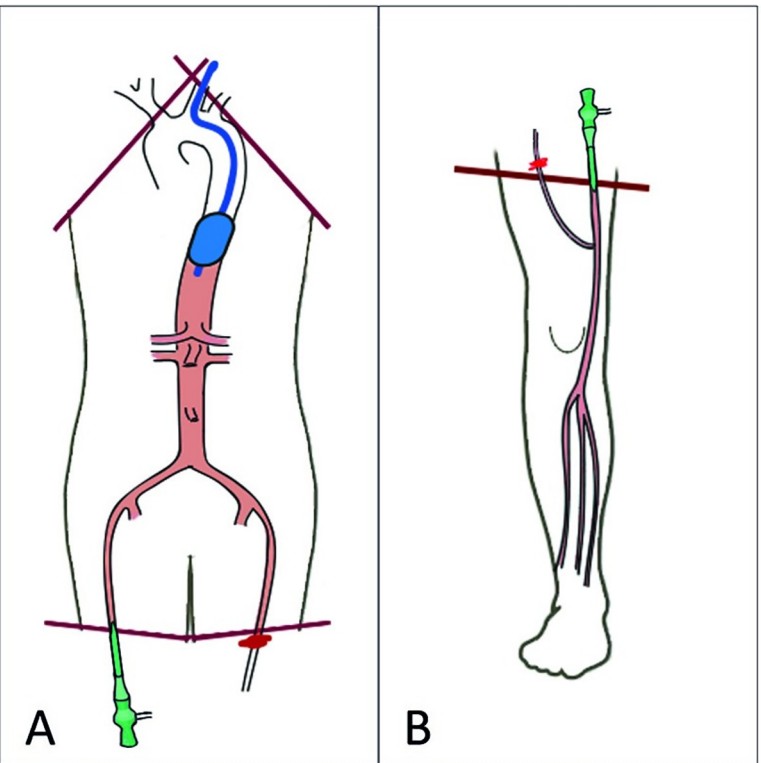

**Fig 1.** Schematic overview of the torso (A) and lower extremity (B) including dissection planes of the cadaveric specimen. Red lines indicate the dissection planes. The access sheath is indicated in green. Fluid leakage is prevented by ligation of the main collateral arteries in the dissection plane, and, in the torsos, by balloon occlusion of the proximal aorta and ligation of the contralateral femoral artery.

and part of the upper leg. An axial dissection plane at mid-femoral level exposed the femoral artery. The dissection planes of the torsos and lower extremities are illustrated in Fig 1.

## Cadaver preparation and storage

The cadaveric specimens were thawed at 15˚C. Approximately one day of thawing was needed for the lower extremities and three days for torsos. The specimens were used for a maximum of four freeze-thaw cycles, due to deterioration of the soft tissues after prolonged use. No difference in quality of the specimen for endovascular use was noticed between first and last use of the specimen.

Vacant arteries in the intersection plane (typically the superficial femoral artery) were cannulated and flushed with a solution of warm tap water and non-oxidative laundry detergent (OMO®, Unilever, London, the UK) until clear liquid was recovered from the venous and arterial outflow. This process reduced odor and resolved post-mortem blood clots. The specimens were flushed with the same solution before restoring them, as low temperatures could cause crystallization of remnant contrast material [16].

## Computed tomography

After several initial experiments, it was decided to subject the specimens were subjected to computed tomography (CT) scans prior to endovascular use. These scans allowed the identification of arterial pathology, calcifications and stenosis to select appropriate specimens for

specific training or research purposes based on vascular pathology, as the medical records of the donors were sealed. The CT scans were acquired with a 64-detector CT scanner (IQon spectral CT, Philips®, Best, The Netherlands) using thin slice, high-dose settings to maximize image quality (slice thickness; 1 mm, spiral pitch; 0.39, tube voltage; 140 kVp, 100–400 mAs).

### Endovascular interventions

The procedures were conducted in an experimental hybrid OR, equipped with a Philips Allura Xper FD20 fluoroscopy system (Philips Healthcare, Best, the Netherlands) and power injector (Mark 7 Arterion, Medrad, Whippany, USA). A vascular surgeon with > 5 years of experience in endovascular interventions performed the endovascular tasks.

**Preparation.**   A 7 Fr introducer sheath was advanced in the femoral artery to provide either retrograde (torsos) or antegrade (lower extremities) arterial access, as shown in Fig 2. The sheath was secured to surrounding tissue with a nylon string or clamp. In all specimens, fluid leakage was prevented by ligation of the main collateral arteries in the dissection plane(s). Additionally, in the torsos, the contralateral femoral artery was clamped and the supratruncal aorta was occluded with a 40x30mm aortic valvuloplasty balloon (Z-MED; NuMED Inc®, Hopkinton). Within these restricted vascular segments, intraluminal pressure was created by injection 50cc (lower extremities) or 100cc (torso) of tap water through the sheath. Fig 1 provides a schematic overview of the setup in the torsos and lower extremities.

**Procedures.**   Various endovascular procedures were performed using this fresh-frozen human cadaver model, both for both training and research purposes. These procedures included balloon angioplasty, stenting, and image optimization of cone beam computed tomography scans (CBCT). Throughout these experiments, the vascular patency of the fresh frozen cadavers was documented by means of cannulation of selected target vessels. These vessels comprised the celiac artery, superior mesenteric artery and left and right renal arteries in the torsos, and the anterior tibial artery, posterior tibial artery and peroneal artery in the lower extremities. These cannulations are comparable to the tasks performed during complex aortic

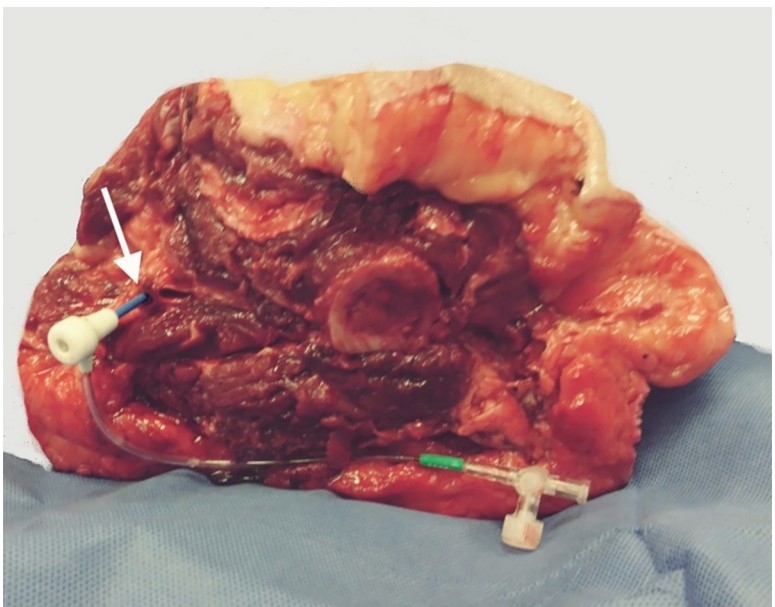

**Fig 2. Cannulation of the femoral artery of a lower extremity.** A 7Fr sheath is used to create antegrade access of the lower extremity through the vacant femoral artery (white arrow) in the axial dissection plane.

interventions and peripheral interventions. In case of cannulation failure, a distinction was made between failure due to flaws in the cadaver model or failure due to steno-occlusive vascular disease.

**Digital subtraction angiography.** Digital subtraction angiography (DSA) was acquired using the following protocols:

*DSA protocol lower extremity*. Manual injection of 50 cc tap water through a Berenstein or Straight flush catheter to generate adequate intraluminal pressure, followed by injection of 20 cc of diluted (ratio 1:5) contrast agent (Ultravist® 300 mg/mL, Bayer Healthcare Pharmaceuticals Inc. Wayne, Germany). Subsequently, 50cc of tap water is injected to dilute and flush the remnant contrast material from the arteries.

*DSA protocol torso*. Manual injection of 100 cc tap water through a pigtail catheter to generate adequate intraluminal pressure, followed by injection of 120 cc diluted (ratio 1:5) contrast agent (Ultravist 300mg/mL) at 20 cc/s using a power injector. Remnant contrast material was flushed with 100 cc of tap water.

## Results

Feasibility of this model is based on our institute's experiences throughout the use of 6 fresh frozen human cadaver torsos and 22 lower extremities.

### Computed tomography

Pre-procedural unenhanced computed tomography (CT) scans were available of 15 of the 22 lower extremities (68%) and 4 of the 6 torsos (66%). The CT scans allowed evaluation of the state of the vascular wall and arterial lumen, as visualized in Fig 3. The two most common port-mortem CT artefacts were decomposition artefacts (e.g. gas formation in the intestine and liver) and artefacts due to anatomical dissection (presence of air within vessels and compression of the aorta, iliac arteries and veins). However, none of these artefacts decreased the diagnostic value of the scan in terms of vascular disease.

### Endovascular interventions

The endovascular human cadaver model provided a realistic simulation setting for a variety of endovascular procedures. DSA imaging was realistic as shown in Fig 4. Air bubbles were visible in 23 of the 92 DSA's (25%), caused by insufficient prefilling of the vasculature (Fig 5). These bubbles had minimal impact on the diagnostic or navigational value of the DSAs, as the outline of the arteries remained visible.

The cadaver model demonstrated a high vascular patency. The celiac artery, superior mesenteric artery, and left and right renal artery could be successfully cannulated in all six torsos (100%). Cannulation of the posterior tibial artery, anterior tibial artery and peroneal artery was successful in 20 (91%), 21 (95%) and 19 (86%) specimens respectively. All six cases of cannulation failure were attributed to high grade stenosis or occlusions of the target vessel, as confirmed by their lack of patency on DSA imaging or, when CT-imaging was available, by presence of local arterial calcifications. More information on these methods can be found in S2 Table.

No visible tissue distortion or swelling of the specimen was observed, even after multiple hours of use. However, venous outflow and sheath leakage were noticeable. In the torsos, partial collapse of the infra-renal aorta was noticed after approximately half an hour, that impaired guidewire manipulation. The full range of guidewire motion could be retrieved by the injection of 100cc of tap water in the restricted arterial segment.

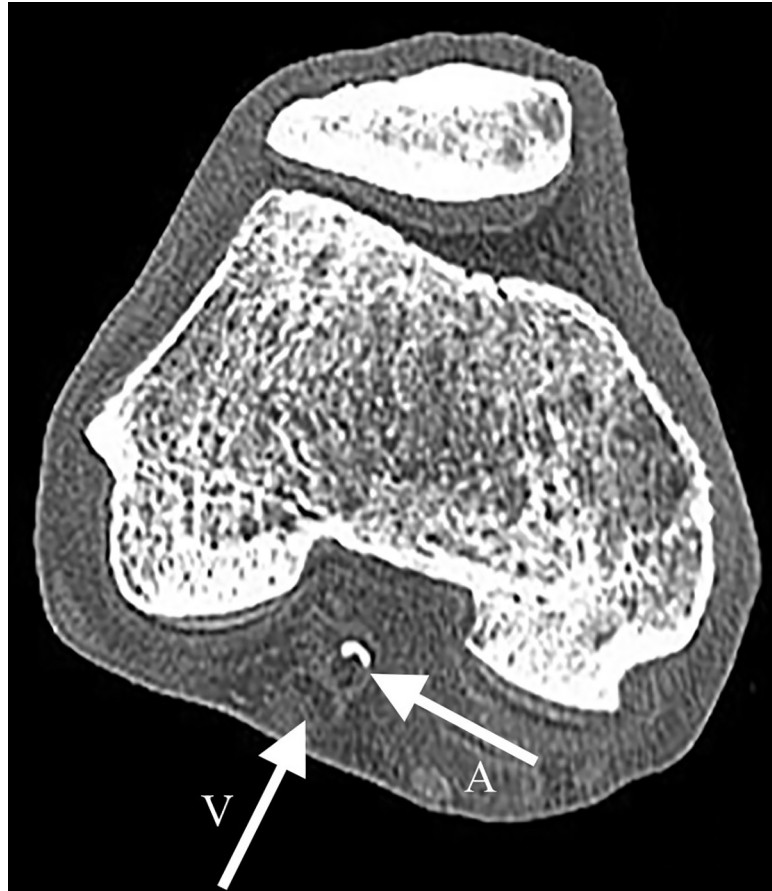

**Fig 3. Unenhanced CT of frozen lower extremity of a human cadaver.** The vein and artery are indicated with an arrow with 'A' and 'V' respectively. Differentiation between arterial lumen, vascular wall and calcifications are possible.

## Discussion

Fresh frozen human cadavers form a suitable model to evaluate new endovascular devices and techniques or master endovascular skills. Human cadavers offer lifelike conditions with representative anatomy, pathophysiology and tactile feedback.

In contrast to previously reported (human) cadaver models [11–14, 17–20], the current model does not employ arterial circulation. Arterial circulation is undeniably relevant for research that concerns hemodynamic processes such as stent deployment, or when maximal expansion of the arteries is desired to allow high-calibre devices. However, many research and training objectives may suffice in a model without circulation, especially since there are several disadvantages to post-mortem reperfusion. Extensive preparation of cadaver and fluid conduit, and an expensive extracorporeal pump are needed to generate stable circulatory flow and prevent conduit leakage. Extravasation of reperfusion fluid into the interstitial space and abdominal cavities forms another issue, as post-mortem decay increases the vascular permeability. Arterial circulation is therefore only sustainable for a limited period of time, and with low blood pressure levels, before resulting in massive edema and distortion of the soft tissues and organs [11–14, 18, 19].

In the current model, arterial circulation was omitted without compromising vessel patency or DSA quality. This allows us to maintain the anatomical and pathophysiolocal fidelity of

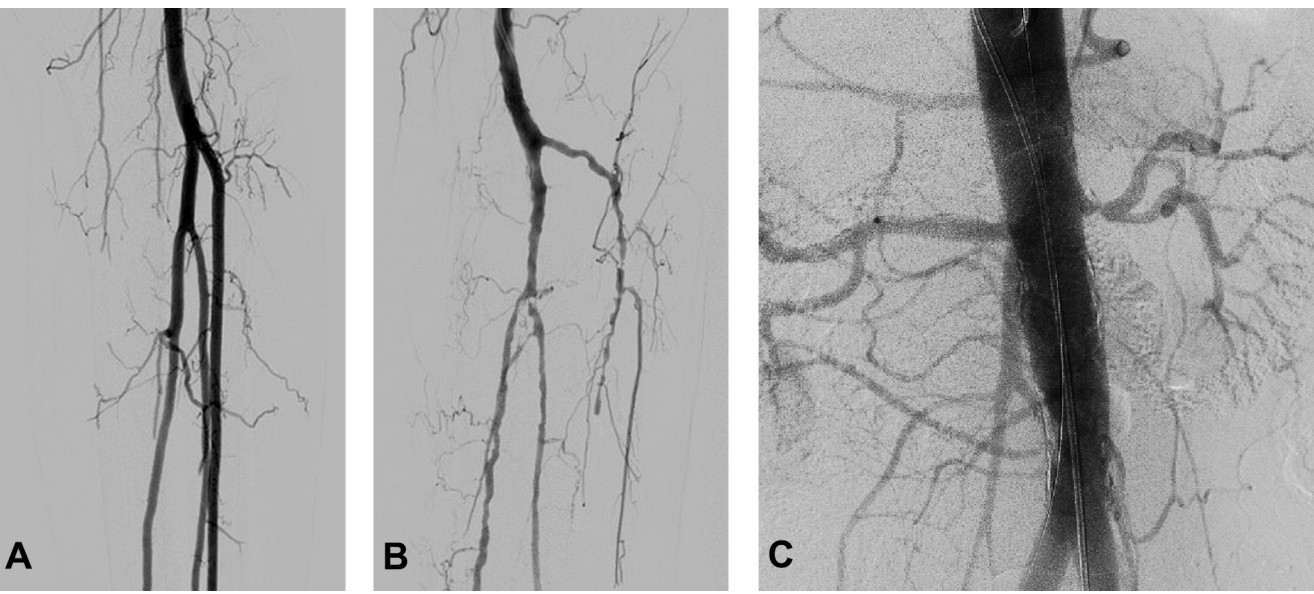

**Fig 4.** DSA imaging of A) the trifurcation and crural arteries of a specimen with patent arteries, B) the trifurcation and crural arteries of a specimen with systemic atherosclerosis and C) the abdominal aorta and its visceral and renal branches.

fresh frozen human cadavers, while using less preparation time, fewer resources and causing significantly less edema than the existing cadaver reperfusion models.

This endovascular human cadaver model has several limitations. To start, the intraluminal pressure was not measured in these specimens. Arterial patency was assumed when guidewires and catheters could be manoeuvred freely, without movement restrictions due to arterial compression. These subjective observations were based on tactile feedback of the devices, and their

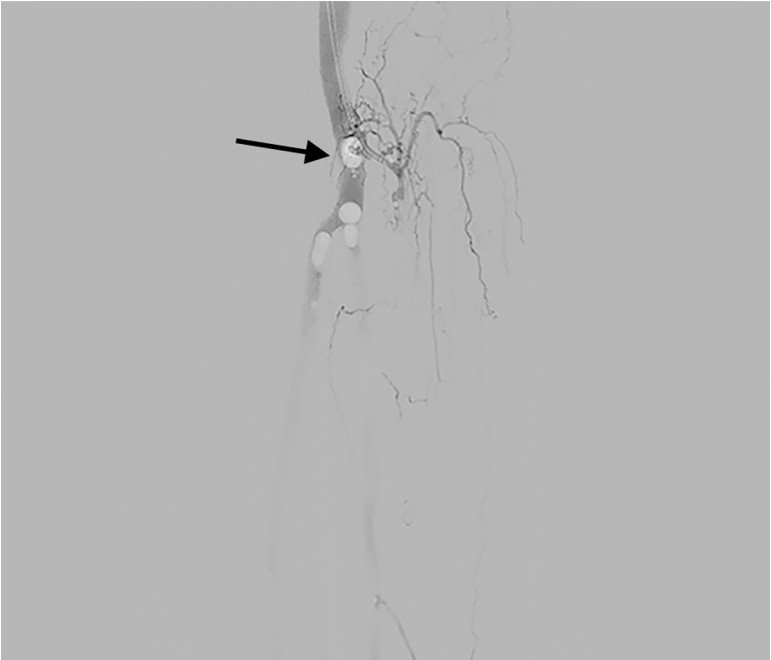

**Fig 5. Air bubbles during angiography.** Presence of air bubbles near the trifurcation, indicated with an arrow.

fluoroscopic feedback. However, maximum expansion of the arteries may not have been obtained, which may limit the use of large calibre devices.

Despite the existence of plenty of thriving body donation programs, it may be difficult to acquire the fresh frozen human cadavers. When available, the cadavers should be used as effectively as possible, as the preservation duration of fresh frozen tissue is limited. Our institution stimulates purposeful use of the cadavers by separating torso, head and extremities for use throughout different medical disciplines.

The specimen used in this study were reused for maximal four times. Chemical body preservation methods could extend the durability of the cadavers. However, fresh frozen preservation is the most lifelike conservation method currently available to preserve flexibility, color and texture of all tissues [21]. Chemical preservation using Thiel's method [22] provides similar levels of tissue flexibility, but may affect the elasticity of the arteries in a similar way as previously reported in tendons and ligaments [23, 24].

Ideally, the vascular anatomy of the cadaveric specimen should match the intended research or training purpose to maximize the fidelity of the endovascular simulation. The privacy of donors is highly respected and therefore our all medical records, including cause of death, are sealed to secure the donors anonymity. However, this policy may be different in other institutions. Without medical records, we were forced to acquire CT scans of the frozen specimen to assess their vascular status. On CT assessment, plenty of lower extremities contained arteries with atherosclerosis and calcifications, however more uncommon pathology may be more difficult to find.

## Conclusions

The endovascular fresh frozen human cadaver model presented in this study allows realistic simulation of endovascular procedures in the aorta and peripheral arteries. This model is suitable for preclinical research and development of endovascular technology, and to gain practical experience and master endovascular skills.

## Supporting information

**S1 Table. Baseline characteristics of body donors.** Gender and age of the donated specimens. (DOCX)

**S2 Table. Target vessel cannulation success.** Table summarizing target vessel cannulation success in all specimens, including information on angiographic visibility of the target vessel and calcification of the target vessel. (DOCX)

## Acknowledgments

We thank the department of anatomy and their employers, who provided insight and expertise that greatly assisted the research, in particular Simon Plomp and Fiona van Zoomeren. Furthermore, we thank the department of experimental cardiology for the ability to use their experimental facility.

## Author Contributions

**Conceptualization:** Marloes M. Jansen, Constantijn E. V. B. Hazenberg, Joost A. van Herwaarden.

**Data curation:** Marloes M. Jansen, Constantijn E. V. B. Hazenberg.

**Investigation:** Marloes M. Jansen, Constantijn E. V. B. Hazenberg, Quirina M. B. de Ruiter, Joost A. van Herwaarden.

**Methodology:** Marloes M. Jansen, Quirina M. B. de Ruiter, Joost A. van Herwaarden.

**Resources:** Ronald L. A. W. Bleys.

**Supervision:** Constantijn E. V. B. Hazenberg, Joost A. van Herwaarden.

**Visualization:** Robbert W. van Hamersvelt.

**Writing – original draft:** Marloes M. Jansen.

**Writing – review & editing:** Constantijn E. V. B. Hazenberg, Quirina M. B. de Ruiter, Joost A. van Herwaarden.

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
