## [Decision Letter · Decision Letter 0]

1 Jul 2020

PONE-D-20-02240

Feasibility of fresh frozen human cadavers as a research and training model for endovascular image guided interventions

PLOS ONE

Dear Dr. Jansen,

Thank you for submitting your manuscript to PLOS ONE. After careful consideration, we feel that it has merit but does not fully meet PLOS ONE’s publication criteria as it currently stands. Therefore, we invite you to submit a revised version of the manuscript that addresses the points raised during the review process.

Please read carefully the comments and suggestions from all three reviewers and fully address each concerns in the revision.

We look forward to receiving your revised manuscript.

Kind regards,

Aloke Finn, MD

Academic Editor

PLOS ONE

Journal Requirements:

2. We note your statement that cadavers were obtained from body donations, according to Dutch law. However, we ask you to amend your methods section to specify from where you obtained the cadavers. If they were donated directly to your institute, please state as such, including the relevant department.

Additional Editor Comments (if provided):

Reviewers' comments:

Reviewer's Responses to Questions

**Comments to the Author**

1. Is the manuscript technically sound, and do the data support the conclusions?

Reviewer #1: Yes

Reviewer #2: Yes

Reviewer #3: Partly

2. Has the statistical analysis been performed appropriately and rigorously? 

Reviewer #1: N/A

Reviewer #2: N/A

Reviewer #3: N/A

3. Have the authors made all data underlying the findings in their manuscript fully available?

Reviewer #1: Yes

Reviewer #2: No

Reviewer #3: Yes

4. Is the manuscript presented in an intelligible fashion and written in standard English?

Reviewer #1: Yes

Reviewer #2: Yes

Reviewer #3: Yes

5. Review Comments to the Author

Reviewer #1: The study is relevant since it addresses the question of finding suitable models for testing of new clinical methods and training - endovascular procedures specifically. There are just a few things that needs to be corrected/clarified in the manus:

Reg. Introduction: It seems like reference no 11 has fallen out here?

Reg. Cadaver preparation and storage: The specimens were used for a maximum of four freeze-thaw cycles. The authors need to describe in short how the increasing deterioration affects the procedures and outcome - is it the cannulation, the injection, the scanning or the results that are affected?

Reg. Digital subtraction angiography: It is described that the injection of tap water is to generate adequate intraluminal pressure. If the pressure was not measured, how was it possible to determine if the inserted volume was sufficient? Also, the sentence "Remnant contrast material was flushed with 50 cc of tap water" is not clearly understandable. This needs to be rephrased or explained in further detail.

Reg: Results: "No edema and third space fluid were observed" - was that observed by the use of scanning? Venous outflow and sheath leakage can be observed by the naked eye, but that might not be the case with edema.

Reviewer #2: Jasen MM et al., reported new method using frozen human cadavers for evaluating results of EVT or training EVT. This study is very unique. However, there are some limitation in this manuscript.

1. As the authors mentioned, this model is to evaluate results of EVT. However, no post-EVT images were provided. Did the lesions show effect of treatment which was supposed to be observed? When those effect are detected and able to be evaluated, we can say this model is established. Therefore, this reviewer would like to know how we can evaluate pre- and post- EVT results using clinical CT. Please provide those information with images or data etc.

2. How did the authors collect donors? Were there any criteria to collect cadavers, e.g., age, DM, HT, smoking? No table showing donor characteristics was provided in this manuscript. Please provide it.

3. Did authors monitor pressure in vessels to pretend actual condition of interventions? Also, what type of pressure did the authors use in this model? Steady flow or pause wave? Do authors think these differences matter to pretend actual clinical settings?

Reviewer #3: To the authors,

In this paper, Jansen et al. proposed a fresh frozen human cadaver model for research and training of endovascular procedure. This model does not need extracorporeal pump and only needs tap water, some sheath introducers, aortic valvuloplasty balloon. The authors performed endovascular procedure including the acquisition of DSA, balloon angioplasty, stenting, and mechanical atherectomy. The author reported no edema and third spacing fluid collection, even after multiple hours of use. The author concluded that their fresh frozen human cadaver model allows realistic simulation of endovascular procedures in the aorta and peripheral arteries and is suitable for preclinical research and development of endovascular technology, and to practice endovascular skills. The cadaver model presented this manuscript seems reasonable, but this manuscript has several concerns as follows.

1. The author performed POBA, stenting, and atherectomy in the cadaver model. But there is no explanation about these procedures in detail. Please describe the results of these procedures in the cadaver model. Which procedure is good or bad for the cadaver model?

2. Experience from our institution, we cannot keep lower limb artery open appropriately without perfusion because of compression of limb muscles. Enough expansion is needed to perform some procedures, such as an atherectomy and intravascular imaging. Please describe the result of atherectomy procedures. If the authors performed, please provide information about intravascular imaging.

3. On page 17, line 21, “The specimen used in this study could be reused a maximum of four times.” How many times did the authors perform procedure per specimen? Please provide this information.

4. Please describe characteristics of the specimens, such as risk factors, cause of death, past history of peripheral artery disease, and the prevalence of stenotic lesions in the cadaver model.

5. On page 15, line 175, “No edema and third spacing fluid were observed, even after multiple hours of use.” This sentence is based on the subjectivity of the author. Please show the data indicating no edema and no third spacing fluid collection.

6. On page 15, line 174, “available CTA or DSA imaging.“ CT ‘angiography’ is an inadequate word because the authors only performed plane CT in the post-mortem CT.

6. PLOS authors have the option to publish the peer review history of their article (what does this mean?). If published, this will include your full peer review and any attached files.

Reviewer #1: **Yes: **Lene Warner Thorup Boel

Reviewer #2: No

Reviewer #3: No

---

## [Author Response · Author response to Decision Letter 0]

11 Aug 2020

[Formatted .docx file with response to the reviewers is attached as an additional file (Response to Reviewers)]

Reviewer #1: 

The study is relevant since it addresses the question of finding suitable models for testing of new clinical methods and training - endovascular procedures specifically. There are just a few things that needs to be corrected/clarified in the manus:

Reg. Introduction: It seems like reference no 11 has fallen out here?

Response: Sorry for this mistake. The error with the references is corrected. 

Response: Corrected ref. numbers in [Line 53 and 56]

Reg. Cadaver preparation and storage: The specimens were used for a maximum of four freeze-thaw cycles. The authors need to describe in short how the increasing deterioration affects the procedures and outcome - is it the cannulation, the injection, the scanning or the results that are affected?

Response: We used the specimens a maximum of 4 times (+ a CT scan with the specimen in frozen condition). At this time, tissue generation began to take a toll; muscular tissues in the upper leg began to deteriorate, and the smell increased. In terms of endovascular use, nothing was noticed; cannulation, injection and results remained similar. In theory, the injected fluids may extravasate faster after deterioration of the arteries, but we noticed no ‘leakage’ of contrast material after angiography. 

As our anatomy department intended to use several of the lower extremities for a dissection class, we decided not to use the specimens more than 4 times, to maintain their usability during the dissection class. We rephrased and elaborated on this in the manuscript.

Alterations in manuscript: elaborated on prolonged use:

“The specimens were used for a maximum of four freeze-thaw cycles, due to deterioration of the soft tissues after prolonged use. No difference in quality of the specimen for endovascular use was noticed between first and last use of the specimen.” [line 96-98]

Reg. Digital subtraction angiography: It is described that the injection of tap water is to generate adequate intraluminal pressure. If the pressure was not measured, how was it possible to determine if the inserted volume was sufficient? Also, the sentence "Remnant contrast material was flushed with 50 cc of tap water" is not clearly understandable. This needs to be rephrased or explained in further detail.

Response: You are right, we used a rather subjective method: we simply injected tap water in the arterial vascular tree to increase the intraluminal pressure to maintain patent traversable arteries. In practice this meant that we were able to freely manoeuvre guidewire and catheters without noticing restriction in movement due to arterial compression (both in terms of tactile feedback, and the ‘smoothness’ of device movement on fluoroscopy). We did not measure the intraluminal pressure and thus, we have no sight on real-time intraluminal pressure, nor on intraluminal pressure decline during use of the model. This could be a future improvement for the use of this model. We added this limitation in the discussion.

Furthermore, we elaborated on the ‘flushing’ of contrast material after angiography. 

Alterations in manuscript: 

• Added section in the discussion: [line 227-232]

“This endovascular human cadaver model has several limitations. To start, the intraluminal pressure was not measured in these specimens. Arterial patency was assumed when guidewires and catheters could be manoeuvred freely, without movement restrictions due to arterial compression. These subjective observations were based on tactile feedback of the devices, and their fluoroscopic feedback. However, maximum expansion of the arteries may not have been obtained, which may limit the use of large calibre devices.”

• Elaborated on ‘flushing of remnant contrast material’: [line 154-155].

“Subsequently, 50cc of tap water is injected to dilute and flush the remnant contrast material from the arteries.”

Reg: Results: "No edema and third space fluid were observed" - was that observed by the use of scanning? Venous outflow and sheath leakage can be observed by the naked eye, but that might not be the case with edema.

Response: You are right, we were unable to perform accurate assessment, as the cadaveric specimens were not dissected, nor scanned directly after their use as cadaver model. We adjusted the text to:

Adjustment in Manuscript: 

“No edema and third spacing fluid were visible tissue distortion, or swelling was observed.” [line 192]

Reviewer #2: 

Jansen MM et al., reported new method using frozen human cadavers for evaluating results of EVT or training EVT. This study is very unique. However, there are some limitation in this manuscript.

1. As the authors mentioned, this model is to evaluate results of EVT. However, no post-EVT images were provided. Did the lesions show effect of treatment which was supposed to be observed? When those effect are detected and able to be evaluated, we can say this model is established. Therefore, this reviewer would like to know how we can evaluate pre- and post- EVT results using clinical CT. Please provide those information with images or data etc.

Response: The main intent of this article was to document our experience and gained knowledge of this ‘non-perfused’ fresh frozen cadaver models. To help peers, and spread awareness of human cadaver models and their feasibility for all sorts of endovascular applications. 

Within our institute we have been using this cadaver model for years for all sorts of endovascular applications, to our own satisfaction. As we could not find a similar methodology in literature, we decided to document our methods and considerations in this manuscript. 

Throughout the last 7 years we have used this human cadaver model a couple of times for a wide range of applications, including: 

• Balloon angioplasty and stenting with novice vascular surgeons in training.

• Target vessel catheterization of visceral, renal and peripheral vessels with vascular surgeons in training.

• Target vessel catheterization to assess the feasibility of a new technique to provide 3D device tracking and navigation.

• Use of new atherectomy devices, followed by vessel excision and visual assessment of damage to the intimal and medial layer of the vessel.

• Optimization of 3D cone beam computed tomography settings for:

o Visualization of the visceral and renal arteries during fenestrated or branched aortic repair.

o Visualization of calcifications in the peripheral arteries to improve guidance during revascularization of chronic total occlusions. 

• Assessment of scatter radiation during target vessel cannulation using different methods of radiation protection (drapes, shields, none).

At the time of execution, there was no intend for publication, and the documentation of the experiments was diverse. Most experiments were focussed on gaining clinical experience with certain tasks or devices, and thus the assessment of the cadaver model was scarcely documented and subjective. Imaging data from the older experiments were lacking.

For the draw-up of this manuscript, we were limited to the well-documented sessions (including imaging), which comprised of these 22 lower extremities and 6 torsos. Moreover, we focussed on the two important simulation qualities that could be assessed objectively:

• Patency of the arteries of the cadaver model � using target vessel cannulation success as a parameter

• Quality of angiography in the cadaver model � assessment of DSA quality

Alterations in manuscript: 

Altered aim of the study: 

• Abstract: [line 17 - 29]

“Objective: To develop describe the feasibility of a fresh frozen human cadaver model for research and training model for endovascular image guided procedures in the aorta and lower extremity. 

Methods: Six fFresh frozen human cadaver torsos and 22 lower extremities were used to construct the endovascular model. were selected for model development and feasibility testing. Endovascular access was acquired by inserting a sheath in the femoral artery. The arterial segment of the specimen was restricted by ligation of collateral arteries and, in the torsos, clamping of the contralateral femoral artery and balloon occlusion of the supratruncal aorta. Tap water was administered through the sheath to create sufficient intraluminal pressure to manipulate devices and acquire digital subtraction angiography (DSA). Endovascular cannulation tasks of the visceral arteries (torso) or the peripheral arteries (lower extremities) were performed to assess the vascular patency of the model. Feasibility of this model is based on our institutes experiences throughout the use of six fresh frozen human cadaver torsos and 22 lower extremities.”

• Introduction: [Line 58 - 62]

“In contrast to these human cadaver reperfusion models, The aim of this study was to develop a human cadaver model without arterial circulation. In this study, we present our human cadaver model, and aim to raise general awareness of the existence and feasibility of human cadaver models for endovascular simulation in the abdominal aorta and peripheral arteries.”

2. How did the authors collect donors? Were there any criteria to collect cadavers, e.g., age, DM, HT, smoking? No table showing donor characteristics was provided in this manuscript. Please provide it.

Response: The donors all came in via the anatomy department of our institute. Written informed consent was obtained from the donors during life. After they passed away, all medical records were sealed to secure the anonymity of the donors. The only demographics that remained accessible were their age, gender, and a serological information (post-mortem tests for HIV, hepatitis A, B etc). We were therefore unable to provide a full table with baseline characteristics. We did, however, add the standard deviation of the donors age. 

The reason why we were unable to provide baseline characteristics is mentioned in line 70-72: “Age, gender and serology report of the donors were provided. Other demographics and medical details were sealed to assure the anonymity of the donors, as according to our institution’s policy.”

Alterations in manuscript: 

• Added the origin of the donors: 

“Body donations were acquired directly by our institute’s anatomy department.” [line 66-67]

• Added standard deviation of the donors age:

“Donors were predominantly male (74%) with a mean age of 74.4 ±.8.8” [line 74-75]

3. Did authors monitor pressure in vessels to pretend actual condition of interventions? Also, what type of pressure did the authors use in this model? Steady flow or pause wave? Do authors think these differences matter to pretend actual clinical settings?

Response: The lack of pressure measurement and the lack of pulsatile arterial blood-flow are drawbacks of this model, compared to the existing cadaver reperfusion models. However, the simplicity of this model is also its strength.

While we do believe that continuous pulsatile flow would provide the most realistic simulation setting, we believe that such a setup would have serious drawbacks in terms of leakage, oedema and third space fluid. The six articles in which reperfusion with continuous pulsatile flow was used, describe arterial pressures ranging between 30mmHg (Willaert et al. 2016) of 80mmHg (Carey et al. 2014).1-6 None of which would be clinically representable, and all of which were accompanied with mentions of rapid extravasation of perfusion fluids and noticeable distortion of solid organs and soft tissues. In some cases, even so severe that the experiment had to be terminated early.6 Moreover, the materials used to recreate pulsatile flow are costly and the necessary preparation of the cadaver is lengthy.

Instead of active, continuous reperfusion, we simply injected tap water in the arterial vascular tree to increase the intraluminal pressure to maintain patent traversable arteries. In practice this meant that we are able to freely manoeuvre guidewire and catheters without noticing restriction in movement due to arterial compression (based on tactile feedback, and on the smoothness of device movements on fluoroscopy). We did not measure the intraluminal pressure and thus, we have no sight on real-time intraluminal pressure, nor on intraluminal pressure decline during use of the model. In our opinion, however, this model functioned properly and we came to believe that the simplicity of this model is actually its strength as it makes the model more accessible for institutions with limited resources. 

1. Willaert W, Tozzi F, Van Hoof T, Ceelen W, Pattyn P, D’Herde K. Lifelike vascular reperfusion of a thiel-embalmed pig model and evaluation as a surgical training tool. Eur Surg Res. 2016;56(3–4):97–108.

2. Carey JN, Minneti M, Leland HA, Demetriades D, Talving P. Perfused fresh cadavers: Method for application to surgical simulation. Am J Surg [Internet]. 2015;210(1):179–87. Available from: http://dx.doi.org/10.1016/j.amjsurg.2014.10.027

3. Chevallier C, Willaert W, Kawa E, Centola M, Steger B, Dirnhofer R, et al. Postmortem circulation: A new model for testing endovascular devices and training clinicians in their use. Clin Anat. 2014;27(4):556–62. 

4. Garrett HE. A human cadaveric circulation model. J Vasc Surg [Internet]. 2001;33(5):1128–30. Available from: http://linkinghub.elsevier.com/retrieve/pii/S0741521401590898

5. Arbatli H, Cikirikcioglu M, Pektok E, Walpoth BH, Fasel J, Kalangos A, et al. Dynamic Human Cadaver Model for Testing the Feasibility of New Endovascular Techniques and Tools. Ann Vasc Surg [Internet]. 2010;24(3):419–22. Available from: http://dx.doi.org/10.1016/j.avsg.2009.04.001

6. Willaert W, De Somer F, Grabherr S, D’Herde K, Pattyn P. Post-mortem Reperfusion of a Pig: a First Step to a New Surgical Training Model? Indian J Surg. 2013;77(December):1–4. 

Adjustments in the manuscript:

• Added a section in the discussion on the lack of pressure measurement: [line 227-232]

“To start, the intraluminal pressure was not measured in these specimens. Arterial patency was assumed when guidewires and catheters could be manoeuvred freely, without movement restrictions due to arterial compression. These subjective observations were based on tactile feedback of the devices, and their fluoroscopic feedback. However, maximum expansion of the arteries may not have been obtained, which may limit the use of large calibre devices.”

Reviewer #3: 

To the authors,

In this paper, Jansen et al. proposed a fresh frozen human cadaver model for research and training of endovascular procedure. This model does not need extracorporeal pump and only needs tap water, some sheath introducers, aortic valvuloplasty balloon. The authors performed endovascular procedure including the acquisition of DSA, balloon angioplasty, stenting, and mechanical atherectomy. The author reported no edema and third spacing fluid collection, even after multiple hours of use. The author concluded that their fresh frozen human cadaver model allows realistic simulation of endovascular procedures in the aorta and peripheral arteries and is suitable for preclinical research and development of endovascular technology, and to practice endovascular skills. The cadaver model presented this manuscript seems reasonable, but this manuscript has several concerns as follows.

1. The author performed POBA, stenting, and atherectomy in the cadaver model. But there is no explanation about these procedures in detail. Please describe the results of these procedures in the cadaver model. Which procedure is good or bad for the cadaver model?

Response: The main intent of this article was to document our experience and gained knowledge of this ‘non-perfused’ fresh frozen cadaver models. To help peers, and spread awareness of human cadaver models and their feasibility for all sorts of endovascular applications.

Within our institute we have been using this cadaver model for years for all sorts of endovascular applications, to our own satisfaction. As we could not find a similar methodology in literature, we decided to document our methods and considerations in this manuscript. 

Throughout the last 7 years we have used this human cadaver model for a wide range of applications, including: 

• Balloon angioplasty and stenting with novice vascular surgeons in training.

• Target vessel catheterization of visceral, renal and peripheral vessels with vascular surgeons in training.

• Target vessel catheterization to assess the feasibility of a new technique to provide 3D device tracking and navigation.

• Use of new atherectomy devices, followed by vessel excision and visual assessment of damage to the intimal and medial layer of the vessel.

• Optimization of 3D cone beam computed tomography settings for:

o Visualization of the visceral and renal arteries during fenestrated or branched aortic repair.

o Visualization of calcifications in the peripheral arteries to improve guidance during revascularization of chronic total occlusions. 

• Assessment of scatter radiation during target vessel cannulation using different methods of radiation protection (drapes, shields, none).

At the time of execution, there was no intend for publication, and the documentation of the experiments was diverse. Most experiments were focussed on gaining clinical experience with certain tasks or devices, and thus the assessment of the cadaver model was scarce and subjective. Imaging data from the older experiments were lacking, due to migration of the experimental imaging database.

For the draw-up of this manuscript, we were limited to the well-documented sessions (including imaging), which comprised of these 22 lower extremities and 6 torsos. Moreover, we focussed on the two important simulation qualities that be established objectively:

• Patency of the arteries of the cadaver model � using target vessel cannulation success as a parameter

• Quality of angiography in the cadaver model � assessment of DSA quality

The issues with subjectivity and lack of proper documentation in the earlier experiments refrained us from specific commentary on the suitability of these devices/procedures in the cadaver model.

Alterations in manuscript: 

Altered aim of the study: 

• Abstract: [line 17 - 29]

“Objective: To develop describe the feasibility of a fresh frozen human cadaver model for research and training model for endovascular image guided procedures in the aorta and lower extremity. 

Methods: Six fFresh frozen human cadaver torsos and 22 lower extremities were used to construct the endovascular model. were selected for model development and feasibility testing. Endovascular access was acquired by inserting a sheath in the femoral artery. The arterial segment of the specimen was restricted by ligation of collateral arteries and, in the torsos, clamping of the contralateral femoral artery and balloon occlusion of the supratruncal aorta. Tap water was administered through the sheath to create sufficient intraluminal pressure to manipulate devices and acquire digital subtraction angiography (DSA). Endovascular cannulation tasks of the visceral arteries (torso) or the peripheral arteries (lower extremities) were performed to assess the vascular patency of the model. Feasibility of this model is based on our institutes experiences throughout the use of six fresh frozen human cadaver torsos and 22 lower extremities.”

• Introduction: [Line 58 - 62]

“In contrast to these human cadaver reperfusion models, our institution has been using human cadaver models without arterial reperfusion, to our satisfaction. The aim of this study was to develop a human cadaver model without arterial circulation. In this study, we present our human cadaver model, and aim to raise general awareness of the existence and feasibility of human cadaver models for endovascular simulation in the abdominal aorta and peripheral arteries.”

2. Experience from our institution, we cannot keep lower limb artery open appropriately without perfusion because of compression of limb muscles. Enough expansion is needed to perform some procedures, such as an atherectomy and intravascular imaging. Please describe the result of atherectomy procedures. If the authors performed, please provide information about intravascular imaging.

Response: Interesting comment! As mentioned, our specimens were dissected at mid femoral level, in which it may be easier to maintain open vessels than in a full body cadaver, which may explain those differences.

After thorough flushing and injection of 50cc of tap water through a sheath in the femoral artery, the arteries and were patent and traversable enough to freely manipulate guidewires and catheters, without noticing arterial compression. However, we must admit that we did not perform a lot of procedures in which high calibre devices were used, which require maximum arterial expansion. 

During our early experiments, we once performed successful atherectomy of a femoral artery using a SilverHawk device (Medtronic). Multiple atherectomy passes were performed, and the femoral artery was later dissected to visually check damage to the intimal and medial layer. However, data collection from this experiment was meagre and relied on subjective experiences as, at that moment, there was no intention for publication. 

We have not used intravascular ultrasound, nor intravascular optical coherence tomography, as we do not have these systems available in our experimental lab, and because we do not frequently use these devices in daily clinical practice.

We understand that readers would like to have more information and accompanying figures to support the use of the atherectomy device. As we are unable to provide detailed information, we deleted the mention of atherectomy from the text. 

Adjustment in Manuscript: 

• Added this as a limitation in the discussion [line 227 -232]

“This endovascular human cadaver model has several limitations. To start, the intraluminal pressure was not measured in these specimens. Arterial patency was assumed when guidewires and catheters could be manoeuvred freely, without noticing movement restriction due to arterial compression. These subjective observations were based on tactile feedback of the devices, and their fluoroscopic feedback. However, maximum expansion of the arteries may not have been obtained, which may limit the use of large calibre devices.”

• Removed the mention of atherectomy, as there is limited objective data/imaging available from this experiment to support this claim. [line 137-138]

“Various endovascular procedures were performed using this fresh-frozen human cadaver model, both for both training and research purposes. These procedures included: balloon angioplasty, stenting and atherectomy and image optimization of cone beam computed tomography scans (CBCT). Throughout these experiments, the vascular patency of the fresh frozen cadavers was documented by means of cannulation of selected target vessels.“

3. On page 17, line 21, “The specimen used in this study could be reused a maximum of four times.” How many times did the authors perform procedure per specimen? Please provide this information.

Response: Actually, we used the specimens a maximum of 4 times (+ a CT scan). At this time, tissue generation began to take a toll; muscular tissues in the upper leg began to deteriorate, as did the smell. As our anatomy department intended to use several of the lower extremities for a dissection class, we decided not to use the specimens more than 4 times, to maintain their use during the dissection class. We rephrased and elaborated on this statement.

Adjustments in Manuscript: 

• Adjusted statement in discussion: [line 238]

“The specimen used in this study were reused for maximal four times.”

• Adjusted Materials and Methods section: [line 96-97]

“The specimens were used for a maximum of four freeze-thaw cycles, due to deterioration of the soft tissues after prolonged use. No difference in quality of the specimen for endovascular use was noticed between first and last use of the specimen.”

4. Please describe characteristics of the specimens, such as risk factors, cause of death, past history of peripheral artery disease, and the prevalence of stenotic lesions in the cadaver model.

Response: The specimens were originated from the anatomy department of our institute. In our institute, all medical records are sealed after body donation, to secure the anonymity of the donors. The only demographics that remained accessible were their age, gender, and a serological information (post-mortem tests for HIV, hepatitis A, B etc). We were therefore unable to provide a full table with baseline characteristics. This is mentioned in line 70-72 (“Age, gender and serology report of the donors were provided. Other demographics and medical details were sealed to assure the anonymity of the donors, as according to our institution’s policy.”)

Adjustments in Manuscript: Added an additional section in the Discussion to put more emphasis on this issue:

“The privacy of donors is highly respected and therefore our all medical records, including cause of death, are sealed to secure the donors anonymity. However, this policy may be different in other institutions. Without medical records, we were forced to acquire CT scans of the frozen specimen to assess their vascular status.” [line 245-248]

5. On page 15, line 175, “No edema and third spacing fluid were observed, even after multiple hours of use.” This sentence is based on the subjectivity of the author. Please show the data indicating no edema and no third spacing fluid collection.

Response: You are right, this is not an accurate assessment, as the cadaveric specimens were not dissected, nor scanned directly after their use as cadaver model. We adjusted the text to:

Adjustment in Manuscript: Adjustment in the Result section:

“No visible tissue distortion, or swelling was observed.” [line 192]

6. On page 15, line 174, “available CTA or DSA imaging.“ CT ‘angiography’ is an inadequate word because the authors only performed plane CT in the post-mortem CT.

Response: Adjusted this sentence

Adjustment in manuscript: [line 188-191]

“All six cases of cannulation failure were attributed to high grade stenosis or occlusions of the target vessel, as confirmed by DSA imaging, or when available, by local arterial calcifications on CT imaging, indicating a chronic total occlusion.”

---

## [Decision Letter · Decision Letter 1]

24 Sep 2020

PONE-D-20-02240R1

Feasibility of fresh frozen human cadavers as a research and training model for endovascular image guided interventions

PLOS ONE

Dear Dr. Jansen,

Thank you for submitting your manuscript to PLOS ONE. After careful consideration, we feel that it has merit but does not fully meet PLOS ONE’s publication criteria as it currently stands. Therefore, we invite you to submit a revised version of the manuscript that addresses the points raised during the review process.

Please correct the manuscript according to the reviewers' suggestions.

We look forward to receiving your revised manuscript.

Kind regards,

Aloke Finn, MD

Academic Editor

PLOS ONE

Reviewers' comments:

Reviewer's Responses to Questions

**Comments to the Author**

1. If the authors have adequately addressed your comments raised in a previous round of review and you feel that this manuscript is now acceptable for publication, you may indicate that here to bypass the “Comments to the Author” section, enter your conflict of interest statement in the “Confidential to Editor” section, and submit your "Accept" recommendation.

Reviewer #1: (No Response)

Reviewer #2: All comments have been addressed

Reviewer #3: All comments have been addressed

2. Is the manuscript technically sound, and do the data support the conclusions?

Reviewer #1: (No Response)

Reviewer #2: Yes

Reviewer #3: Yes

3. Has the statistical analysis been performed appropriately and rigorously? 

Reviewer #1: (No Response)

Reviewer #2: Yes

Reviewer #3: Yes

4. Have the authors made all data underlying the findings in their manuscript fully available?

Reviewer #1: (No Response)

Reviewer #2: Yes

Reviewer #3: Yes

5. Is the manuscript presented in an intelligible fashion and written in standard English?

Reviewer #1: (No Response)

Reviewer #2: Yes

Reviewer #3: Yes

6. Review Comments to the Author

Reviewer #1: (No Response)

Reviewer #2: This reviewer does not have further comments. The manuscript was corrected based on our suggestions and questions.

Reviewer #3: The manuscript was well revised and I have only minor comments as follows.

1. Regarding the supporting information, the excel files should be summarized using tables, not raw data (e.g., how many cases were documented as ‘Angiographically visible’ etc…). Also, please provide the abbreviations list for the supporting information.

2. I think if supporting information would be refined, it would be nice information for the readers. Please define the definitions for ‘Angiographically visible’ and ‘calcification visible’.

7. PLOS authors have the option to publish the peer review history of their article (what does this mean?). If published, this will include your full peer review and any attached files.

Reviewer #1: **Yes: **Lene Warner Thorup Boel

Reviewer #2: No

Reviewer #3: No

---

## [Author Response · Author response to Decision Letter 1]

29 Sep 2020

Dear editor and reviewers,

Thank you for your comments based upon the first revision of the manuscript PONE-D-20-02240. In this second revision we have thoroughly revised the supporting information, as elaborated on in the comments below. 

We look forward to hearing from you regarding this submission. We would be glad to respond to any further questions and comments that you may have.

On behalf of all authors,

Marloes Jansen, MSc.

PhD-candidate Vascular Surgery

Review Comments to the Author:

Reviewer #2: This reviewer does not have further comments. The manuscript was corrected based on our suggestions and questions.

Response: Thank you for your suggestions.

Reviewer #3: The manuscript was well revised and I have only minor comments as follows.

1. Regarding the supporting information, the excel files should be summarized using tables, not raw data (e.g., how many cases were documented as ‘Angiographically visible’ etc…). Also, please provide the abbreviations list for the supporting information.

2. I think if supporting information would be refined, it would be nice information for the readers. Please define the definitions for ‘Angiographically visible’ and ‘calcification visible’.

Response: Thank you for your suggestions. Based on your comments we have revised the supporting information files. Firstly, we deposited the underlying (raw) data at a public data depository (Mendeley Data) as suggested in Plos One’s publication guidelines. The data is accessible with the DOI provided in the reference section. [line 262-263 of the redline document].

Secondly, both supporting information files (S1 and S2) now contain summarization tables, rather than raw data. Both files contain more elaborate background information including a table legend, spelled out abbreviations and definitions. 

We reframed the term ‘angiographically visible’ to ‘patent on angiogram’. And provided definitions of the terms ‘patent on angiogram’ and ‘visible calcifications’ in the tables. 

Moreover, we have included several example images, to provide a more visual understanding of these definitions. These images include illustration(s) of 1) ‘target vessel cannulation success’, 2) ‘patent on angiogram’ and 3) ‘calcification visible’. [S1 Tables and S2 Tables]

To convey this information to the readers, we referred to the S2 table in the manuscript. [line 183 – 188 of the redline document]

“Cannulation of the posterior tibial artery, anterior tibial artery and peroneal artery was successful in 20 (91%), 21 (95%) and 19 (86%) specimens respectively. All six cases of cannulation failure were attributed to high grade stenosis or occlusions of the target vessel, as confirmed by their lack of patency on DSA imaging or, when CT-imaging was available, by presence of local arterial calcifications. More information on these methods can be found in Supporting Information S2.”

---

## [Decision Letter · Decision Letter 2]

6 Nov 2020

Feasibility of fresh frozen human cadavers as a research and training model for endovascular image guided interventions

PONE-D-20-02240R2

Dear Dr. Jansen,

We’re pleased to inform you that your manuscript has been judged scientifically suitable for publication and will be formally accepted for publication once it meets all outstanding technical requirements.

Kind regards,

Aloke Finn, MD

Academic Editor

PLOS ONE

Additional Editor Comments (optional):

Reviewers' comments:

Reviewer's Responses to Questions

**Comments to the Author**

1. If the authors have adequately addressed your comments raised in a previous round of review and you feel that this manuscript is now acceptable for publication, you may indicate that here to bypass the “Comments to the Author” section, enter your conflict of interest statement in the “Confidential to Editor” section, and submit your "Accept" recommendation.

Reviewer #1: All comments have been addressed

Reviewer #3: All comments have been addressed

2. Is the manuscript technically sound, and do the data support the conclusions?

Reviewer #1: Yes

Reviewer #3: Yes

3. Has the statistical analysis been performed appropriately and rigorously? 

Reviewer #1: N/A

Reviewer #3: Yes

4. Have the authors made all data underlying the findings in their manuscript fully available?

Reviewer #1: Yes

Reviewer #3: Yes

5. Is the manuscript presented in an intelligible fashion and written in standard English?

Reviewer #1: Yes

Reviewer #3: Yes

6. Review Comments to the Author

Reviewer #1: I have no further comments to the paper after this second review. All the issues addressed by the reviewers have been met.

Reviewer #3: The manuscript has been revised well. The author addressed reviewers comments and made the tables based on the reviewer's suggestion.

7. PLOS authors have the option to publish the peer review history of their article (what does this mean?). If published, this will include your full peer review and any attached files.

Reviewer #1: **Yes: **Lene Warner Thorup Boel

Reviewer #3: No

---

## [Editor Report · Acceptance letter]

16 Nov 2020

PONE-D-20-02240R2 

Feasibility of fresh frozen human cadavers as a research and training model for endovascular image guided interventions 

Dear Dr. Jansen:

I'm pleased to inform you that your manuscript has been deemed suitable for publication in PLOS ONE. Congratulations! Your manuscript is now with our production department. 

Kind regards, 

on behalf of

Dr. Aloke Finn 

Academic Editor

PLOS ONE